# Control of Explosive Chemical Reactions by Optical Excitations: Defect-Induced Decomposition of Trinitrotoluene at Metal Oxide Surfaces

**DOI:** 10.3390/molecules28030953

**Published:** 2023-01-18

**Authors:** Roman V. Tsyshevsky, Sergey N. Rashkeev, Maija M. Kuklja

**Affiliations:** Department of Materials Science and Engineering, University of Maryland, College Park, MD 20742, USA

**Keywords:** high energy density materials, decomposition mechanisms, surface and interfacial defects, electronic excitation, photocatalysis

## Abstract

Interfaces formed by high energy density materials and metal oxides present intriguing new opportunities for a large set of novel applications that depend on the control of the energy release and initiation of explosive chemical reactions. We studied the role of structural defects at a MgO surface in the modification of electronic and optical properties of the energetic material TNT (2-methyl-1,3,5-trinitrobenzene, also known as trinitrotoluene, C_7_H_5_N_3_O_6_) deposited at the surface. Using density functional theory (DFT)-based solid-state periodic calculations with hybrid density functionals, we show how the control of chemical explosive reactions can be achieved by tuning the electronic structure of energetic compound at an interface with oxides. The presence of defects at the oxide surface, such as steps, kinks, corners, and oxygen vacancies, significantly affects interfacial properties and modifies electronic spectra and charge transfer dynamics between the oxide surface and adsorbed energetic material. As a result, the electronic and optical properties of trinitrotoluene, mixed with an inorganic material (thus forming a composite), can be manipulated with high precision by interactions between TNT and the inorganic material at composite interfaces, namely, by charge transfer and band alignment. Also, the electron charge transfer between TNT and MgO surface reduces the decomposition barriers of the energetic material. In particular, it is shown that surface structural defects are critically important in the photodecomposition processes. These results open new possibilities for the rather precise control over the decomposition initiation mechanisms in energetic materials by optical excitations.

## 1. Introduction

Energy release processes in molecular energetic materials [1], while not fully understood, are of great interest to scientists and engineers because of their wide array of applications, ranging from medicine to rocket engine fuels and environmental sciences. These applications include the design of new materials [2] with targeted superior properties (e.g., superhard and superdense materials) [3,4], the discovery of new pharmaceuticals [5,6] for medicine, the development of safe and efficient explosives for construction and mining industries [7], the improvement of explosive sensors and detectors for combat against terrorism [8,9], and innovations in the handling and recycling of toxic explosives for sustainable and clean environment [10].

In this work, we reveal the role of interfacial defects at a MgO surface in modifications of the electronic structure and optical spectra of adsorbed energetic trinitrotoluene nitro compound. Such a situation may occur when an explosive material is mixed with an oxide, forming a composite with multiple interfaces [11,12,13,14,15,16]. Surface and interfacial defects [17,18,19] play critically important roles in many chemical and biological processes. These include, for instance, reactions at surfaces and interfaces of microchips used in electronics, and reactions in heterogeneous catalysis used for cleaning emissions in automobile exhausts. Changes in the electronic structure and optical properties observed at interfaces between organic materials and metal oxides open new prospects in the fields of photovoltaics and photochemistry.

Although some composites were used as efficient visible light photocatalysts [20,21], the information about the physical and chemical properties of composites that include energetic materials is very limited. Interfaces between high energy density materials and oxides, and, especially, chemical reactions triggered by surface and interfacial defects, are no exceptions and largely represent an uncharted territory. In particular, it is well-known that the sensitivity of an energetic material could significantly change when this material is mixed with another material, i.e., an explosive reaction could be modified by the presence of interfaces. Such a phenomenon is similar to what one observes in heterogeneous catalysis, where the chemical reactions can be controlled by interfaces. However, the information about electronic properties of interfaces between solid oxides and energetic materials that are related to the interaction with light is sparse.

To design energetic materials (EM) with low sensitivity (stable materials which are difficult to explode) and high performance (able to release a lot of thermal energy), one needs to achieve the control over initiation of chemistry in molecular EM at metal oxide surfaces, i.e., to understand the factors that govern chemical decomposition at these surfaces (surface chemistry, presence of specific types of surface defects, etc.) [18,19] and design surfaces that favor the desired type of reactions [22,23]. Once a better understanding of these processes is reached, a significant number of new applications will become available, including a detection of explosives at low concentrations (an important problem in contemporary antiterrorism efforts) and a design of processes for removing explosives from the environment (for example, TNT is poisonous and may cause skin irritation and harmful effects on the immune system).

To solve this problem, one should identify a suitable chemical process that facilitates the control of energy release. It was shown that the activation barrier of decomposition of different energetic molecules could be significantly lowered for charged and excited states of the molecules [24,25,26]. Moreover, it was found that the excited states can not only change the energetics of chemical reactions by reducing the reaction barriers, but also change the dominating decomposition chemical pathway [26]. The decomposition mechanism may be also affected by the binding of the EM molecule to some moiety [24] and by the polarity of energetic molecular crystal surfaces [23]. Also, the decomposition of EM molecules, positioned at a surface of inorganic solid material (oxide, perovskite, etc.), may significantly differ from the decomposition in vacuum conditions [12,13,14,15,16].

These facts indicate that some control over energy release from energetic materials may be achieved by two steps: (i) positioning EM at solid inorganic material surfaces (or mixing EM with other material to form a composite containing multiple interfaces) and (ii) optically exciting the system to initiate the decomposition process. The electronic and optical properties of EM molecules positioned at an interface could be significantly affected by interfacial interactions, such as charge transfer and band alignment. Typically, electronic orbitals of the adsorbed molecule are hybridized with those of the inorganic surface, which generates a number of local electronic states in the bandgap of the original inorganic material (e.g., oxide, perovskite) that can be optically excited by a specific light frequency. Specifically, it was shown that when TNT molecules are adsorbed at a surface of the well-known catalyst TiO_2_ above a surface oxygen vacancy, 0.62 electron is transferred from the surface to the molecule [13]. The effect of charge transfer is to make TNT molecules negatively charged, and the C-N bond cleavage energy decreases significantly from 76 kcal/mol to only 52 kcal/mol.

Also, for PETN-MgO and TNT-MgO composite samples, it was proven (both theoretically and experimentally) that chemical decomposition reactions could be controlled by electronic transitions at EM–oxide interfaces and optically stimulated by laser light [14,15]. Recently, the role of F^0^-centers at an MgO surface in decomposition of energetic nitro compounds PETN, RDX, and TNT has been investigated [27]. It was found that the first step of the molecular decomposition is initiated by an electron charge transfer from one of the F^0^-centers at the surface to the NO_2_ group of the energetic molecule, which results in the cleavage of one of the N-O bonds. The O atom released from the nitro molecule goes to the oxygen vacancy, thus liquidating the vacancy defect. This decomposition initiation mechanism is significantly different from regular decomposition in the gas phase or solid state for all three considered nitro energetic molecules. In particular, the activation barriers for N-O cleavage are much lower than the activation barriers for the first step of decomposition reactions in gas phase or solid state (molecular crystals) for all three nitro materials.

This work presents the first report to consider the modification of the electronic structure and optical spectra of energetic material trinitrotoluene, adsorbed at structural and coordination defects in MgO surface including steps, kinks, and corners, and so the calculations take into account a realistic (non-ideal) microstructure of the oxide surface. We chose TNT as a good representative of the C-nitro energetic materials that show higher stability (higher decomposition energy barrier) than widely used nitramines, including RDX (1,3,5-Trinitroperhydro-1,3,5-triazine, C_3_H_6_N_6_O_6_) and HMX (1,3,5,7-Tetranitro-1,3,5,7-tetrazocane, C_4_H_8_N_8_O_8_), and nitro esters, including PETN (Pentaerythritol tetranitrate, C_5_H_8_N_4_O_12_) [28]. Our study shows that the presence of undercoordinated sites at the MgO surface has a significant effect on the electronic structure and optical properties of the adsorbed TNT molecule. We argue that tunability of the optical properties of TNT-MgO composites could be efficiently used for the control of chemical explosion reactions in these systems, as well as for environmentally friendly TNT handling and recycling. Our theoretical calculations and predictions are compared with available experimental measurements of optical absorption in TNT-MgO composites.

## 2. Details of Calculations

Solid-state periodic calculations were performed by employing density functional theory [29,30] (DFT) with vDW-DF [31,32,33] functional of Langreth, Lundqvist *et al.,* which includes corrections of van der Waals interactions, as implemented in the VASP code [34,35,36]. To correct the significantly underestimated band gap energies, obtained from vDW-DF, a self-consistent single-point calculation was performed for each configuration by using hybrid PBE0 functional [37]. The projector-augmented wave (PAW) approximation [38] was used.

In calculations of an ideal MgO crystal, the convergence criterion for total energy was set to 10^−5^ eV, and the maximum force acting on each atom in the periodic cell was set not to exceed 0.01 eV/Å. We used 3 × 3 × 3 Monkhorst−Pack ***k***-point mesh, and the kinetic energy cut-off was set to 520 eV. The calculated MgO lattice constant of the cubic (rock salt) unit cell, ***a*** = 4.250 Å, agrees with the experimental lattice vectors of ***a*** = 4.216 Å [39] within 1%.

To simulate the MgO surface, we used a periodic slab model. A rectangular block of 8×8×4 ions represented a supercell, which consists of the four-layer MgO slab cut from the bulk MgO structure to form the MgO surface with the (001) orientation, with the supercell lattice vectors of ***a*** = ***b*** = 17.0 Å and ***c*** = 24.38 Å. The vacuum layer of 20 Å, placed on top of the (001) MgO surface, was intended to minimize interactions between supercells in the ***z*** direction and to ensure that the electronic states of different slabs do not overlap.

To simulate the MgO surface with structural defects (a kink and step), we used a periodic slab model consisting of 10 × 10 × 4 ions with one layer added to simulate structural defects. The supercell lattice vectors of the system are ***a*** = ***b*** = 21.25 Å and ***c*** = 28.5 Å. The kinetic energy cut-off was reduced to 400 eV.

For a TNT-MgO composite, we used a simplified model that consists of a TNT molecule at a MgO surface in a large supercell, instead of constructing an interface between MgO and solid TNT (molecular crystal). Such a model is well justified because: (i) there are no significant interactions between molecules in a molecular crystal; (ii) for a TNT molecule adsorbed at MgO surface, its interactions with the surface are much stronger than with the neighboring TNT molecules in molecular crystals (which are mostly defined by van der Waals forces). Therefore, one can neglect the neighboring TNT molecules without losing the quality of description of important interactions in the system.

All *molecular* calculations in our study were carried out within the GAUSSIAN 09 [40] program suite. Equilibrium ground state structures and energy gaps between the highest occupied molecular orbital (HOMO) and lowest unoccupied molecular orbital (LUMO) were obtained using PBE0 functional with 6-31 + G(d,p) basis set. Vertical excitation energies for the lowest singlet (S) and triplet (T) states were computed using the time-dependent TD PBE0 method [41,42].

## 3. Electronic and Optical Properties of TNT, MgO, and TNT-MgO Interfaces

### 3.1. Molecular and Crystalline TNT

The molecular and crystal structures of TNT are depicted in Figure 1a,b. The highest occupied molecular orbital (HOMO) of an isolated TNT molecule is predominantly formed from 2*p*_z_ atomic orbitals (*z* is the direction perpendicular to the plane of the molecule) of aromatic ring carbon and oxygen atoms (Figure 1c). The lowest unoccupied molecular orbital (LUMO) is formed by 2*p*_z_ atomic functions of nitrogen and oxygen atoms from the nitro groups and carbon atoms from the aromatic ring (Figure 1d).

The calculated energy gap of a TNT bulk ideal crystal is 4.86 eV (Figure 1e). Previous calculations [13] indicate that this energy is close to the energy of the lowest singlet–singlet S_0_→S_1_ transition in a single molecule, calculated using TD DFT method. This is not surprising because TNT molecules in the molecular crystal do not interact strongly. The top of the valence band and the bottom of the conduction band of the TNT molecular crystal are also mainly formed by the 2*p*- orbitals of oxygen and nitrogen atoms, which similar to the HOMO and LUMO orbitals of an isolated molecule. The plot of the frequency-dependent imaginary part of the dielectric function (ε_2_(ω)), corresponding to the optical absorption spectrum of bulk TNT crystal, is shown in Figure 1f. This curve has a maximum at 5.16 eV.

### 3.2. Pristine and Defect-Containing MgO Surface

Magnesium oxide is a well-known wide gap insulator with a band gap of 7.78 eV [39,43]. The MgO (001) surface, however, has a noticeably lower energy gap of 5.5–6.15 eV [44]. When low coordinated sites such as steps, kinks, and oxygen vacancies (Figure 2) are present at the surface, optical transitions of lower excitation energies, 4.5–5.5 eV [45,46,47,48,49] appear. Additionally, at rough MgO surfaces and oxide powders, transitions with photon energies as low as 3.5 eV were observed [27].

According to our calculations, the electronic structure of MgO is characterized by the strong localization of valence electrons and a large band gap between occupied anion (O *2p*) and unoccupied cation (Mg *3s*) states. The calculated energy gap of the ideal bulk MgO crystal is 7.20 eV (Figure 3a, Table 1), which is in reasonable agreement with the experimental value of 7.78 eV [12,13]. The plot of the frequency-dependent imaginary part of the dielectric function (Figure 3b), corresponding to optical absorption spectrum of MgO ideal bulk, has a distinct maximum at 7.4 eV, which also satisfactorily agrees with experiment [13].

The calculated band gap of the MgO (001) surface is reduced to 5.55 eV, relative to the bulk value of 7.20 eV (Figure 3a, Table 1), which is in agreement with the high-resolution electron energy loss spectra of MgO that have demonstrated the exciton absorption peak at 5.5 eV [14]. The calculated optical absorption spectrum of MgO (001) surface has two distinct maximums, at 5.57 and 7.30 eV (Figure 3c), which correspond to the absorption at the surface and in the bulk, respectively.

We also performed calculations for step, kink, and F^0^-center defects at the (001) MgO surface (Figure 2). All three defects generate additional electronic states in the energy gap of the MgO (Figure 3a). The monoatomic step does not produce any noticeable effect on the electronic structure of MgO, as may be seen from Figure 3a. The calculated energy gap of the (001) MgO surface containing a monoatomic step (5.43 eV) is only ~0.1 eV lower than the energy gap of the pristine surface (5.55 eV), which is consistent with the earlier theoretical study [50].

The other two defects affect the electronic structure of the MgO surface much more strongly.

Thus, the kink generates a localized state in the MgO energy gap (Figure 3a), which is predominantly formed by 2*p* states of the three-coordinated oxygen and lies 1.36 eV above the top of the valence band; this is consistent with the previous estimation of 1.2 eV [51]. The calculated HOMO–LUMO gap of the MgO surface containing the kink with a three-coordinated oxygen atom is 4.19 eV, which is consistent with experiment on desorption of oxygen atoms from under-coordinated surface states under optical excitation with the energy of 4.7 eV [52].

Besides the kink containing a three-coordinated oxygen atom, we also considered a kink terminated with three-coordinated Mg atom. The latter defect generates an unoccupied local state in the energy gap of MgO, which is formed by 3*s* states of three-coordinated Mg atom and 2*p* states of the neighboring four-coordinated oxygen atoms. This state lies 1.96 eV below the bottom of the conduction band (Figure 3a). As a result, the HOMO–LUMO gap of the system containing neighboring kinks with three-coordinated oxygen and three-coordinated magnesium ions is 2.23 eV (Table 1, Figure 3a), which is consistent with the low energy peak of 2.3 eV observed in electron energy loss spectrum of MgO [53].

The F^0^-center at the (001) MgO surface generates localized electronic states in the band gap at 3.10 eV above the top of the valence band and 2.45 eV below the bottom of the conduction band (Figure 3a). An optical absorption spectrum calculated for MgO surface with F^0^-center (Figure 3d) shows four distinct peaks at 3.10, 4.53, 5.51 and 7.23 eV. The two latter peaks correspond to the absorption in the ideal, defect-free MgO bulk crystal and at its surface, whereas peaks at 3.10 and 4.53 eV are associated with absorption of the F^0^-center. According to the reflectance spectrum, MgO powder contains an absorption band at 4.58 eV [17,18], which tends to disappear rapidly with powder sintering. The absorption spectrum of MgO crystals irradiated by neutrons has adsorption bands of the 4.95, 3.49 and 1.27 eV, the intensity of which tends to increase with higher neutron dose [54]. On the other hand, an electron energy-loss spectroscopy study of thermally generated defects in MgO (100) films indicate three distinct peaks, observed at 1.15, 3.58, and 5.33 eV [55]. Our calculations confirm that the experimental absorption bands at energies above 3 eV could be related to the absorption at F^0^- centers.

### 3.3. TNT at MgO (001) Surface

First, we studied several different configurations that consist of a single TNT molecule adsorbed at a pristine (001) MgO surface (Figure 4) with the aim of revealing the most feasible model corresponding to the highest binding energy (E_b_). Our calculations showed that **Model 5**, with the plane of a TNT molecule lying nearly parallel to the oxide surface, corresponds to the highest adsorption energy, E_b_ = 30.1 kcal/mol (Figure 4e). Based on these results, we employed such a configuration (first, the plane of the TNT molecule is taken parallel to the surface, then the structure is fully relaxed) for the exploration of the electronic and optical properties of the TNT molecule adsorbed at both a pristine (001) MgO surface as well as at surfaces containing a monoatomic step, kink and F^0^-center (Figure 5a–d). We note that the adsorption energy of TNT on F^0^-center is 50 kcal/mol (obtained from PBE DFT). Note, this energy is higher than the energy of TNT adsorbed on the pristine surface, meaning that the binding of TNT with the F^0^-center is stronger than that with the ideal surface. 

A TNT molecule, adsorbed at the pristine (001) MgO surface (Figure 5a) as well as at surfaces containing the monoatomic step (Figure 5b) and kink (Figure 5c), generates unoccupied states in the band gap of MgO (Figure 5e–g). These are mainly formed by 2*p* orbitals of oxygen, nitrogen and ring carbon atoms of the TNT molecule similar to the nature of the molecular LUMO. As a result, the HOMO–LUMO gap of the systems containing a TNT molecule adsorbed on the pristine (001) MgO surface is calculated at 2.65 eV, which is ~3 eV lower than the MgO surface energy gap (Figure 5e and Figure 6, Table 1).

The monoatomic step does not have any noticeable effect on the electronic structure of the TNT-MgO system (Figure 5b,f), whereas the calculated HOMO–LUMO gap is only 0.02 eV lower as compared the TNT molecule adsorbed on pristine (001) MgO surface (Figure 5e,f and Figure 6, Table 1).

The kink at MgO surface (Figure 5c,g) has a more pronounced effect on the electronic structure of the MgO-TNT system. Density of states (Figure 5g) reveals several additional electronic states in the bandgap of the MgO surface. The highest occupied state is formed by 2*p* atomic orbitals of the three-coordinated surface oxygen atom, whereas the lowest unoccupied state is localized at oxygen and nitrogen atoms of the TNT molecule, as was mentioned earlier. The calculated HOMO–LUMO gap of the system is 1.73 eV (Figure 5g and Figure 6, Table 1).

The F^0^-center defect strongly affects the geometry and electronic structure of the TNT–MgO system (Figure 5d,h). Bader charges reveal a strong electronic charge transfer from the vacancy to the TNT molecule. The calculated Bader charge siting at the TNT molecule is negative and is equal to 1.9 charge of an electron. Table 1 and DOS (Figure 5h) show that the HOMO–LUMO gap of the system containing TNT molecule adsorbed above the F^0^-center is 1.85 eV, whereas the HOMO state is localized at the MgO surface and the LUMO state is localized mainly at the TNT molecule (see also Ref [15]).

For the system of a TNT molecule adsorbed atop of a kink defect at MgO (001) surface, the calculated HOMO–LUMO band gap is 1.73 eV (Figure 5g and Figure 6). Figure 7 shows charge densities for the highest occupied and lowest unoccupied states of this system. It is apparent that the HOMO state is mainly localized at the oxygen atom sitting at the corner of the kink at MgO surface, while the LUMO state is primarily formed by orbitals of atoms localized at the TNT molecule. There is also a significant delocalization of the LUMO state around the molecule. These density plots indicate that the lowest energy electronic transition in this system results in a significant charge transfer from the surface defect to the adsorbed TNT molecule, i.e., the molecule becomes more electronegative when the system is optically excited.

### 3.4. Optical Absorption at TNT-MgO Interfaces: Comparison to Available Experiments

The optical absorption spectra shown in Figure 8 indicate new adsorption features for a TNT molecule adsorbed at the ideal (001) MgO surface in the region of 2.5–3.5 eV. For a pristine MgO surface, we considered two configurations—**Model 1** and **Model 5** (Figure 4). **Model 1** is considered because: (i) it has one of the highest binding energies among all configurations with the TNT molecular plane perpendicular to the MgO surface; (ii) it has the lowest bandgap among all of such configurations. **Model 2** has nearly the same binding energy but a higher optical bandgap. **Model 5** is considered because: (i) it has the highest binding energy between the molecule and MgO surface; (ii) the plane of the molecule is parallel to the MgO surface, which makes this configuration different from other considered structures. Note that absorption spectra calculated for different configurations of a TNT molecule adsorbed at a pristine (001) MgO surface demonstrate different optical features. Thus, the absorption spectrum of the system corresponding to **Model 1** (Figure 4a and Figure 8) has a distinct peak at ~2.7 eV, whereas the absorption spectrum of the system corresponding to **Model 5** (Figure 4e and Figure 8) has a shoulder in the energy range from 2.8 to 4 eV. This means that, although the main features of the absorption curve survive with the change of an orientation of the adsorbed molecule, some fine details of the spectra may vary. The absorption spectrum of a TNT molecule positioned atop of an F^0^-center at (001) MgO surface shows an intense peak with a distinct maximum at 1.76 eV.

An accurate comparison between the calculations and available experimental data would suggest an interpretation of all fine details of the spectra on the basis of electronic structure of the system that contains: (i) a defect at the MgO surface and (ii) a TNT molecule attached to this defect or positioned in its vicinity.

A detailed classification of coordination defects at the MgO (001) surface, which included steps, corners, and kinks, was performed in Ref. [51]. The results demonstrated the existence of deep and shallow electron traps at these defects and established direct correlation between surface features and their spectroscopic and other electronic properties. In particular, the ionization energies, electron affinities, optical excitation energies, and relaxed electron and hole states at the corners, kinks, and steps of the MgO (001) surface were calculated. All considered defects have electronic states within the bandgap of the pristine MgO (001) surface, i.e., they may be optically excited at energies lower than the oxide bandgap (see also Figure 3 and Table 1). This is in agreement with our results.

When a TNT molecule is attached to one of these defects or is placed in its vicinity, the electronic spectrum of the system becomes significantly more complicated. First, as we discussed above, the TNT (as well as most of other energetic materials molecules with complex stoichiometry) molecule may absorb in different ways depending on its orientation relative to the surface, i.e., the system defect plus molecule form configurations with different electronic spectra (compare, e.g., absorption spectra for **Model 1** and **Model 5**, shown in Figure 8). Second, the existence of electron traps at the surface defects should affect the mechanisms of surface charging and photoinduced surface processes. In particular, these traps may serve as transient states for charge transfer between the defect and TNT molecule electron trapping creating electron-hole pairs.

Therefore, we do not expect direct quantitative comparison of calculations with available experiments to be capable of identifying all the fine details of the absorption spectra for energetic molecules atop of surface defects. However, our calculations show that the main features of the absorption curve survive with the change of a molecule orientation. Therefore, some qualitative analysis of experimental absorption spectra based on electronic structure calculations is possible.

There exist just several experiments on the stimulation of chemical decomposition reactions in energetic materials by laser excitation. In particular, for PETN-MgO and TNT-MgO composite samples, it was proven (both theoretically and experimentally by laser light excitation) that chemical decomposition reactions can be controlled by electronic transitions at EM–oxide interfaces and optically stimulated by laser light [15]. It was found that first (1064 nm, 1.17 eV), second (532, 2.33 eV), and third (355 nm, 3.49 eV) laser harmonics radiation of a YAG: Nd laser can be effectively used to trigger explosive reactions in the PETN-MgO samples. Both pure MgO and TNT materials are completely transparent to all of these light frequencies. Using a combination of DFT periodic calculations, embedded cluster method modeling, and optical absorption spectra measurements it was shown that PETN-MgO composite interfaces absorb light with energies much lower than the fundamental band gap of either the energetic material or MgO. Absorption of low-energy photons by the PETN-MgO engineered interfaces becomes possible due to distinct electronic states localized at PETN molecules and situated 3–3.7 eV above the valence band maximum.

For TNT-MgO composite, similar electronic states are positioned at 2–2.6 eV above the valence band maximum. Both calculations and optical absorption spectra measurements suggested that TNT-MgO interfaces will also absorb sub-band photons through nonlinear optics effects. However, laser initiation experiments did not trigger a decomposition in the entire TNT-MgO sample, an outcome which is probably related to low-energy laser initiation. Namely, because of remarkable stability of TNT molecules, TNT crystals, and TNT anion radicals, interfaces with MgO have little effect on reducing the threshold of initiation for TNT as much higher laser irradiation intensity is required.

Diffuse reflectance spectra of MgO-TNT composites were measured in Refs. [15,56]. In these experiments, magnesium oxide powder was mixed with the TNT energetic material by subjecting it to grinding in an agate mortar. Concentrations of the EM in a mixture varied from 0.2 to several mass percent. The MgO-TNT mixtures were then heated in a drying cabinet above the melting point of the energetic material for 10 min to achieve a better coating of MgO particles by an energetic material. The obtained diffuse reflection spectra were analyzed by using the Kubelka–Munk formula.

Absorption spectra of the MgO-TNT composites have several distinct bands (Figure 9, Table 2). The first maximum at 1.7 eV cannot be explained by an absorption at pure a TNT or defect-free MgO surface. We suggest that it corresponds to an absorption by TNT molecules adsorbed either atop or F^0^-center or at a kink on an (001) MgO surface (Figure 8). The second absorption band with a maximum at 2.6 eV and the shoulder in the region 3.4 eV may correspond to an adsorption by TNT molecules positioned at pristine, defect-free MgO surface. The most intense band with the maximum at ~5.2 eV most likely corresponds to an absorption at a pristine MgO surface.

Therefore, we conclude that optical absorption experiments in MgO-TNT composite samples reveal new distinct absorption bands, which were not observed in the reflection spectra of the individual materials. The reflection spectra of MgO-TNT samples show two new absorption bands at 1.7 and 2.6 eV. These findings are in good agreement with the results of the DFT-based calculations and provide an additional strong support to the conclusions obtained from the computational modeling.

## 4. Conclusions

Summing up, we investigated the role of interfacial defects at an MgO surface during modification of the electronic structure and optical spectra of an adsorbed energetic trinitrotoluene nitro compound. We found that electronic and optical properties of TNT mixed with an inorganic material (thus forming a composite) are dramatically affected by interactions at composite interfaces, namely, by charge transfer and band alignment. In particular, the electron charge transfer between TNT and MgO surface may significantly modify the decomposition barriers of the energetic material. It is shown that structural surface defects should be critically important in the photodecomposition. Changes in the electronic structure and optical properties observed, at interfaces between organic materials and metal oxides, open new prospects in the fields of photovoltaics and photochemistry. Promising applications include, for example, the design of new materials with targeted superior hardness and density, new pharmaceuticals for medicine, the development of safe and efficient explosive devices for construction and mining industries, improvement of explosive sensors and detectors for combat against terrorism, and innovations in the handling and recycling of toxic explosives. Interfaces between high energy density (or energetic) materials and oxides, and especially chemical reactions triggered by surface and interfacial defects, are largely under-investigated and remain outstanding challenges. The tunability of the electronic structure of energetic compounds offers a powerful yet cost-efficient means for the control of chemical explosion reactions in these materials, as well as for the environmentally friendly handling and recycling of these toxic materials. These results open new possibilities for control over the decomposition initiation mechanisms in energetic materials via the use of optical excitations.

## Figures and Tables

**Figure 1 molecules-28-00953-f001:**
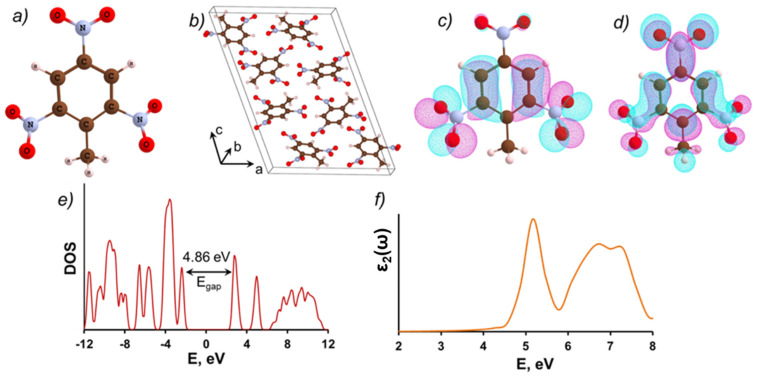
(**a**) The molecular and (**b**) crystalline TNT structures; the electron density distribution of the molecular TNT (**c**) HOMO and (**d**) LUMO orbitals (the isosurfaces corresponding to the coefficients with a positive sign are colored in magenta, and with a negative sign are colored in cyan); (**e**) the total density of states (DOS) of the TNT bulk crystal; (**f**) the frequency-dependent imaginary part of the dielectric function (ε_2_(ω)) of the TNT bulk crystal, averaged over the three crystallographic directions. Carbon atoms are shown in brown, O-in red, N-in blue, H-in white.

**Figure 2 molecules-28-00953-f002:**
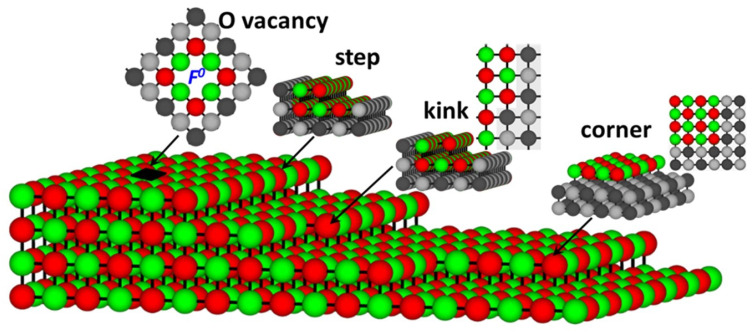
Schematics of the several coordination defects at (001) MgO surface: O vacancy, step, kink, and corner. Mg—in green, O—in red. Atoms that do not belong to the top surface layer are shown in black (oxygen) and gray (magnesium).

**Figure 3 molecules-28-00953-f003:**
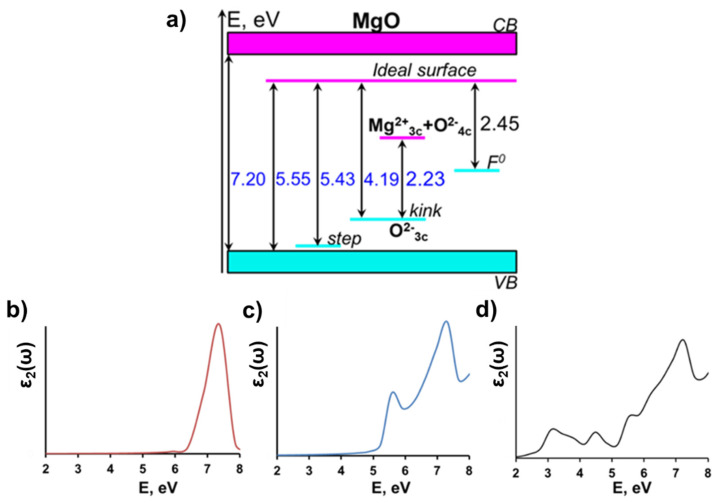
(**a**) A diagram showing the relative energies of local electronic states corresponding to the step, kink, and F^0^-center defects at the MgO (001) surface versus the defect-free bulk and surface magnesium oxide band gaps; the frequency-dependent imaginary part of the dielectric function (averaged over three crystallographic directions) of: (**b**) MgO ideal bulk; (**c**) MgO (001) surface; (**d**) F^0^-center at the MgO (001) surface.

**Figure 4 molecules-28-00953-f004:**
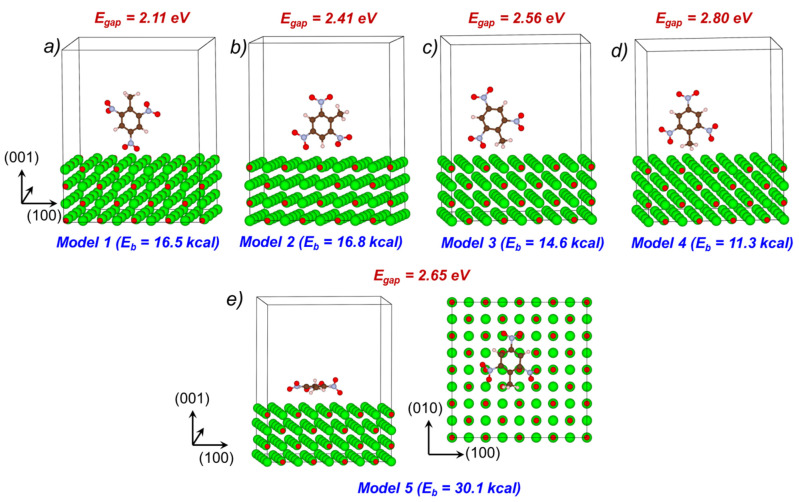
Several slab models with different orientation of a single TNT molecule adsorbed at a pristine (001) MgO surface. For each configuration, the binding energy (*E_b_*) and the bandgap (*E_gap_*) are shown. Mg—in green, O—in red, C—in brown, N—in blue, H—in white.

**Figure 5 molecules-28-00953-f005:**
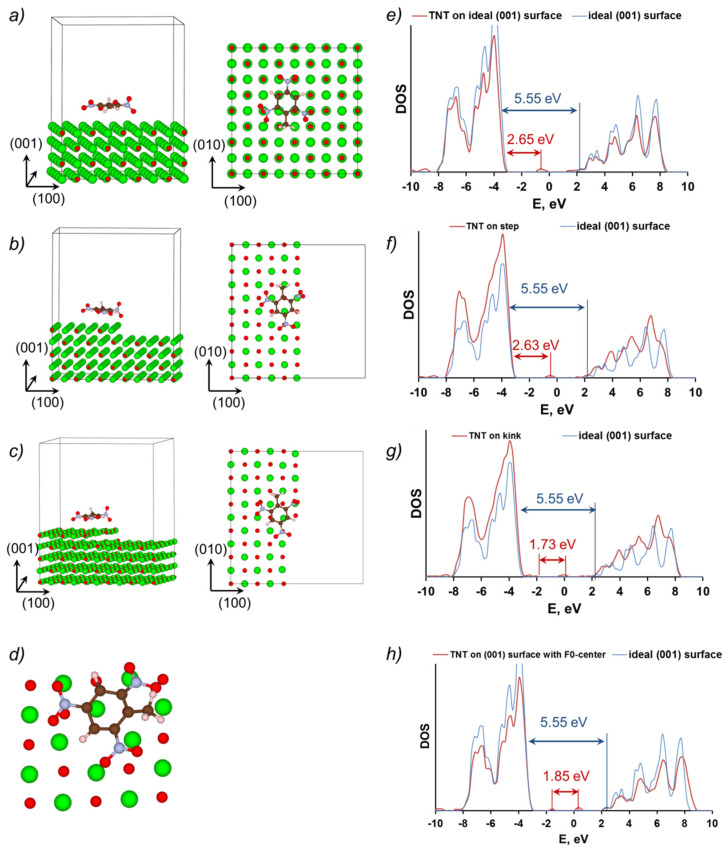
Schematics of the supercells that consist of a TNT molecule adsorbed at: (**a**) a pristine (001) MgO surface; (**b**) a monoatomic step at (001) MgO surface; (**c**) a kink at (001) MgO surface, (**d**) an F^0^-center at (001) MgO surface. Figure also shows total DOS for same configurations: (**e**) a pristine (001) MgO surface; (**f**) a monoatomic step at (001) MgO surface; (**g**) a kink at (001) MgO surface; (**h**) an F^0^-center at (001) MgO surface. Mg—green, O—in red, C—in brown, N—in blue, H—in white.

**Figure 6 molecules-28-00953-f006:**
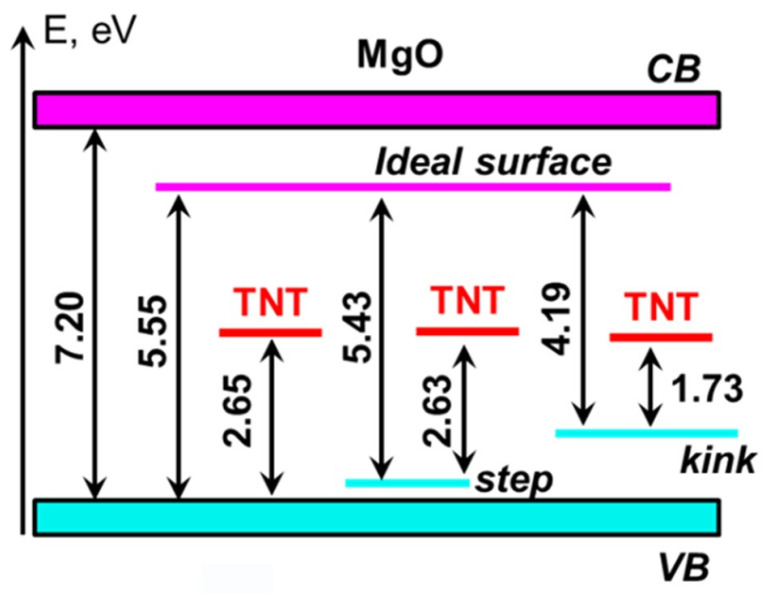
A diagram showing the relative energies of localized electronic states corresponding to the TNT molecule atop of the step and kink defects at the MgO (001) surface versus the defect-free bulk and surface MgO (001) band gaps.

**Figure 7 molecules-28-00953-f007:**
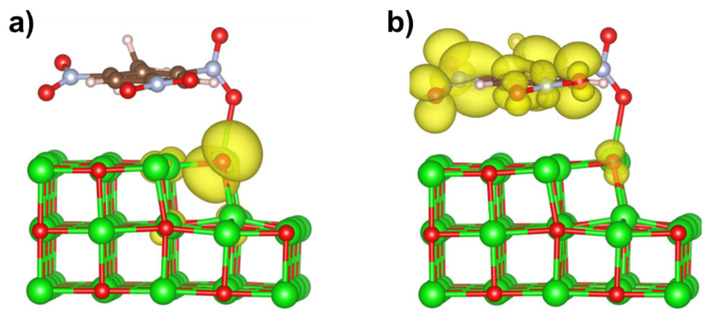
Charge density (shown in three-dimensional yellow contours) for TNT molecule atop a kink at (001) MgO surface (see also Figure 5c) for: (**a**) the highest occupied state; (**b**) the lowest unoccupied state. Mg—in green, O—in red, C—in brown, N—in blue, H—in white.

**Figure 8 molecules-28-00953-f008:**
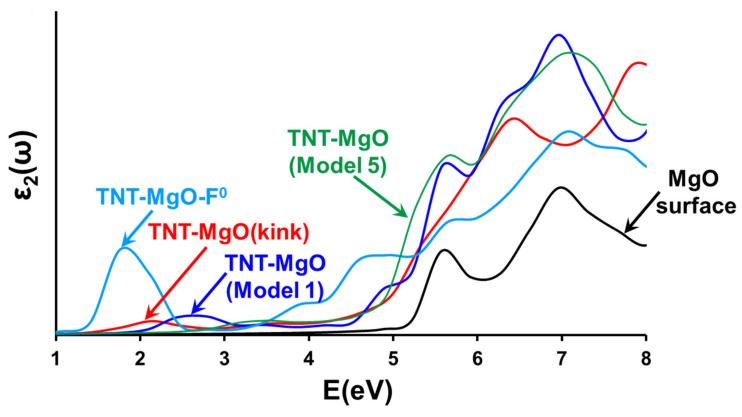
The frequency-dependent, imaginary part of the dielectric function (averaged over the three crystallographic directions) for: a clean (001) MgO surface (black line); a TNT molecule adsorbed at an ideal (001) MgO surface (blue and green lines correspond to **Model 1** and **Model 5** shown in Figure 4); a TNT molecule adsorbed atop of a kink defect (red line); a TNT molecule adsorbed on top of F^0^-center at an (001) MgO surface (cyan line).

**Figure 9 molecules-28-00953-f009:**
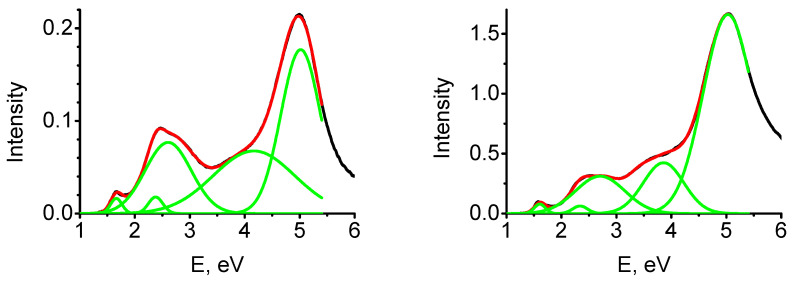
The Kubelka–Munk function (red) and the contributions of different absorption bands (green) for TNT–MgO composites with TNT concentrations of 0.27% (**left**) and 2.5% (**right**).

**Table 1 molecules-28-00953-t001:** Calculated band gaps E_gap_ (eV) for (001) MgO surface (clean and with defects) and for individual TNT molecule adsorbed at the same surfaces.

Model	E_gap_(eV)
**MgO**	
Ideal Bulk Crystal	7.20
Ideal (001) Surface	5.55
Step on (001) Surface	5.43
Kink on (001) Surface	2.23
F^0^- center on (001) Surface	2.45
**TNT Molecule at MgO Surface**	
Ideal (001) MgO Surface	2.65
(001) MgO Surface with Step	2.63
(001) MgO Surface with Kink	1.73
(001) MgO Surface with F^0^ Center	1.85

**Table 2 molecules-28-00953-t002:** Intensities for different absorption bands in the Kubelka–Munk function for TNT–MgO composites with TNT concentrations of 0.27% and 2.5%.

C (%)\Band	1.6 eV	2.3 eV	2.6 eV	3.9 eV	5.0 eV
0.27	0.00427	0.00585	0.08075	0.12555	0.15986
2.5	0.02209	0.01185	0.26136	0.60573	1.79894

## Data Availability

Not applicable.

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
