# Peer review of "Control of Explosive Chemical Reactions by Optical Excitations: Defect-Induced Decomposition of Trinitrotoluene at Metal Oxide Surfaces"

_molecules, 2023, doi:10.3390/molecules28030953_

Round 1
Reviewer 1 Report
The present work investigates the role of structural defects at a MgO surface in the modification of energetic TNT material deposited for electronic and optical properties at the surface. The work is very relevant in developing photovoltaics and photochemistry applications.
My suggestion for the abstract: Please include a sentence that indicates DFT /simulation method used.
In the Introduction section: Surface and interfacial defects are very important for understanding defects and related mechanisms. Defect-trapping or annealing of defects at the interface is important for understanding the chemistry. I suggest including some references in the second paragraph of the intro. After reading this I realized it can be shifted end of the introduction before the result section.
Reference style: Mix-somewhere superscript and somewhere inline
Overall, the work is well-written and well-outlined. However, some points need special care. First, there is a need to introduce a paragraph in the Introduction that emphasizes the novelty of the present work.
Author Response
To Reviewer #1:
The present work investigates the role of structural defects at a MgO surface in the modification of energetic TNT material deposited for electronic and optical properties at the surface. The work is very relevant in developing photovoltaics and photochemistry applications. Overall, the work is well-written and well-outlined.
- My suggestion for the abstract: Please include a sentence that indicates DFT /simulation method used.
Response: We included a short sentence in the abstract describing the method used.
- In the Introduction section: Surface and interfacial defects are very important for understanding defects and related mechanisms. Defect-trapping or annealing of defects at the interface is important for understanding the chemistry. I suggest including some references in the second paragraph of the intro. After reading this I realized it can be shifted end of the introduction before the result section.
Response: We are not sure why the reviewer suggests moving something in the intro section. However, we added relevant references in the introduction.
- Reference style: Mix-somewhere superscript and somewhere inline.
Response: We went through the manuscript to check that all references are inserted in the right format.
- However, some points need special care. First, there is a need to introduce a paragraph in the Introduction that emphasizes the novelty of the present work.
Response: We modified the Introduction section to highlight the novelty of this work.
Reviewer 2 Report
Dear authors,
The TNT is decomposed over the MgO surface and changes observed in the surface electronic structure of metal oxides is discussed. The charge densities of TNT at different sites of MgO surface is the interesting aspect to note with. The listed comments may be considered to further improve our understanding.
Minor comments
Abstract
(1) Specify the crystalline phase of MgO.
(2) The nature of the defects, charge transfer dynamics and the exact changes in the surface electronic structure of MgO observed should be elegantly expressed.
The significant findings of the present work should be conveyed to persuade the readers to follow the full article.
Results and discussion
(1) Figure 4: The adsorption of TNT on the defective site of MgO should be computed. The role of defects in the alteration of adsorption energy (if any) may be discussed.
(2) Is it possible to include some discussions on trinitrobenzene decomposition to understand the influence of –CH3 group of TNT?

Author Response
To Reviewer #2:
The significant changes in the surface electronic structure of MgO attained after the decomposition of TNT on its surface is excellently presented by DFT calculations. The narrowing of the gap region, distinct electronic transitions and orientation of TNT to drive the adsorption process are the other interesting aspects. The article is well-organized and discussions are clear without ambiguity.
Abstract
- Specify the crystalline phase of MgO.
Response: In the original text of the manuscript, we stated that “The calculated MgO lattice constant of the cubic (rock salt) unit cell, a= 4.250 Å, agrees with the experimental lattice vectors of a=4.216 Å” in the Section 2 (page 4), i.e., the crystalline phase of MgO was specified. We consider this answers the question of the reviewer and we feel that there is no need to specify this structure once again in the Abstract.
- The nature of the defects, charge transfer dynamics and the exact changes in the surface electronic structure of MgO observed should be elegantly expressed. The significant findings of the present work should be conveyed to persuade the readers to follow the full article.
Response: We modified the Abstract to highlight the nature of defects, the charge transfer mechanisms, and modifications of the electronic structure at the interface.
Results and discussion
- Figure 4: The adsorption of TNT on the defective site of MgO should be computed. The role of defects in the alteration of adsorption energy (if any) may be discussed.
Response: We added the adsorption energy of TNT on F-center in the Results section. It now reads: “We note that the adsorption energy of TNT on F0-center is 50 kcal/mol (obtained from PBE DFT). Reasonably, the energy is higher than the energy of TNT adsorbed on the pristine surface, meaning that the binding of TNT with the F0-center is stronger than with the ideal surface.”
- Is it possible to include some discussions on trinitrobenzene decomposition to understand the influence of –CH3 group of TNT?
Response: In this work, we focus on the initiation of the decomposition based on the interaction between the energetic material and solid oxide surface with defects rather than considering all the details of TNT decomposition. The initiation of chemistry starts with the detachment of the NO2 group from a TNT molecule. The role of the -CH3 group comes later, while in the initiation process it is negligible and, hence, we did not consider it here. We feel that the discussion on the full decomposition, including secondary reactions, (such as -CH3) would detract the attention of the reader from the main conclusions and our findings.
Reviewer 3 Report
In this article " Defect-Induced Decomposition of Trinitrotoluene at Metal Oxide Surfaces ", The author mentioned about the Decomposition of Trinitrotoluene at Metal Oxide Surfaces. After reviewing this article, Overall, the quality looks good however, there are some issues that have to be concern before consideration it for publication;
Please provide any symbol to the end of author's name and at begining of the affiliation at supescript position.
The title should be revised to somewhat catchy type.
The abstract could be more specific towards the outcome results of this study.
Please revised introduction with proper consequences with some new references by exploring the literature and specify the novelty of the study. (Journal of Molecular Structure 1098, 393-399, Applied Nanoscience 11 (4), 1291-1302).
why the author only consider model 1 and 5 please specify in better way.
Some errors regarding the sub/super script, spacing and typo need to consider throughout the manuscript. (like most of the ref mentioned in text such as 44, 45 49 and % on the top of figure 9)
Make sure that the format of references are uniform.
In the conclusion author mention that this material will be suitable to application however did not mention application.
Author Response
To Reviewer #3:
- Please provide any symbol to the end of author's name and at begining of the affiliation at supescript position.
Response: We did not provide any symbol to the end of each author’s name because all the authors have the same affiliation, we are at the University of Maryland College Park. Such symbols would make sense if affiliations were different, which is not the case.
- The title should be revised to somewhat catchy type.
Response: We agree with the reviewer that the title may be more intriguing and hence changed it to “Control of Explosive Chemical Reactions by Optical Excitations: Defect-Induced Decomposition of Trinitrotoluene at Metal Oxide Surfaces “.
- The abstract could be more specific towards the outcome results of this study.
Response: We revised the Abstract to specify the nature of defects, the charge transfer mechanisms, and modifications of the electronic structure at the interface.
- Please revised introduction with proper consequences with some new references by exploring the literature and specify the novelty of the study. (Journal of Molecular Structure 1098, 393-399, Applied Nanoscience 11 (4), 1291-1302).
Response: We added references in the introduction section where we discuss the novelty of the paper.
- why the author only consider model 1 and 5 please specify in better way.
Response: Model 1 is considered because: (i) it has one of the highest binding energies for all configurations with the TNT molecule plane perpendicular to the MgO surface and (ii) is has the lowest bandgap for all of such configurations. Model 2 has nearly the same binding energy but higher optical bandgap. Model 5 is considered because: (i) it has the highest binding energy between the molecule and MgO surface and (ii) the plane of the molecule is parallel to the MgO surface, which makes this configuration different from other considered structures. We added this explanation to the text (see page 12).
- Some errors regarding the sub/super script, spacing and typo need to consider throughout the manuscript. (like most of the ref mentioned in text such as 44, 45 49 and % on the top of figure 9). Make sure that the format of references are uniform.
Response: We checked the text for consistence and accuracy of formatting.
- In the conclusion author mention that this material will be suitable to application however did not mention application.
Response: We added a sentence to the Conclusion section (see page 16) with the list of promising applications.